# Determination of Free Amino Acids in Milk, Colostrum and Plasma of Swine via Liquid Chromatography with Fluorescence and UV Detection

**DOI:** 10.3390/molecules27134153

**Published:** 2022-06-28

**Authors:** Roberto Gotti, Erika Esposito, Diana Luise, Stefano Tullio, Nicolò Interino, Paolo Trevisi, Jessica Fiori

**Affiliations:** 1Department of Pharmacy and Biotechnology FaBit, Alma Mater Studiorum, University of Bologna, Via Belmeloro 6, 40126 Bologna, Italy; roberto.gotti@unibo.it; 2Department of Chemistry “G. Ciamician”, Alma Mater Studiorum, University of Bologna, Via F. Selmi 2, 40126 Bologna, Italy; erika.esposito3@studio.unibo.it (E.E.); stefano.tullio2@studio.unibo.it (S.T.); nicolo.interino2@unibo.it (N.I.); 3Department of Agricultural and Food Sciences, Alma Mater Studiorum, University of Bologna, Viale G. Fanin, 40127 Bologna, Italy; diana.luise2@unibo.it (D.L.); paolo.trevisi@unibo.it (P.T.)

**Keywords:** free amino acids, milk, colostrum, plasma, derivatization, liquid chromatography, swine

## Abstract

Amino acids are ubiquitous components of mammalian milk and greatly contribute to its nutritional value. The compositional analysis of free amino acids is poorly reported in the literature even though their determination in the biological fluids of livestock animals is necessary to establish possible nutritional interventions. In the present study, the free amino acid profiles in mature swine milk, colostrum and plasma were assessed using a targeted metabolomics approach. In particular, 20 amino acids were identified and quantified via two alternative and complementary reversed-phase HPLC methods, involving two stationary phases based on core-shell technology, i.e., Kinetex C18 and Kinetex F5, and two detection systems, i.e., a diode array detector (DAD) and a fluorescence detector (FLD). The sample preparation involved a de-proteinization step, followed by pre-chromatographic derivatization with 9-fluorenylmethylchloroformate (FMOC-Cl). The two optimized methods were validated for specificity, linearity, sensitivity, matrix effect, accuracy and precision and the analytical performances were compared. The analytical methods proved to be suitable for free amino acid profiling in different matrices with high sensitivity and specificity. The correlations among amino acid levels in different biological fluids can be useful for the evaluation of physio-pathological status and to monitor the effects of therapeutic or nutritional interventions in humans and animals.

## 1. Introduction

The analysis of free amino acids (FAAs) and protein-bounded amino acids (protein-AAs) in foods, biological fluids and tissues is of great importance for nutritional, biochemical and physiological studies, and has been extensively applied for the diagnosis and monitoring of inherited metabolic disorders. In milk, FAAs have many different biological functions beyond protein synthesis, as they play a central role in the metabolism, growth and development of mammals. Changes in FAA level can be attributed to nutrition, environmental factors or genetic modulation and can be related to altered states of metabolism [1]. Nutrition studies have demonstrated that dietary FAA supplementation modulates gene expression, improves small intestine and skeletal muscle growth and reduces excess body fat [2,3].

Due to their important biological functions, AAs are often used as dietary food supplements, not only for humans, but also and above all for livestock animals. Nevertheless, despite the wide use of FAAs in the livestock diet as supported by the extensive literature regarding their positive effects on the performance and profitability of the animal, only a few studies have addressed the compositional analysis of FAAs in the matrices, including the blood plasma and milk, of swine [3,4,5].

In blood, plasma and urine, FAA concentration can be a direct consequence of the intake of protein or of AAs themselves. Bertocchi et al. [6] observed that a shortage of dietary leucine (Leu) in a pig’s diet was associated with the increase of numerous AAs in its blood. Leu is a key signal for muscle deposition as its deficiency can impair body protein synthesis, provoking a loss of AAs [7]. On the other hand, the FAA profiles of colostrum and milk are of interest due to the lack of information on the fate of AAs in the mammary gland [8]. It is known that the uptake of some AAs is not directly related to their concentration in milk, due to specific pathways for catabolism or biosynthesis. Moreover, the FAA profiles of colostrum and milk are of utmost interest because they can be related to the modulation of the piglet’s growth [9]. The punctual detection of the FAA profile in these matrices may contribute to the study of metabolomics regulation and dysregulation as well as the study of the biological mechanism in which AAs are involved and, in turn, help nutritionists design specific nutritional interventions [6,10].

Different from the chromatographic compositional analysis of protein-AAs, which is carried out upon the acidic or enzymatic hydrolysis of the matrix, FAA determination requires higher sensitivity because of the lower content of analytes and higher selectivity since detection occurs in untreated matrices often containing high fat levels and interfering compounds (e.g., in mature milk and colostrum). 

Different analytical methods that address the quantitation of FAAs have been developed over the years; ion-exchange chromatography (IEC) with post-column ninhydrin derivatization coupled with UV detection, defined as the “AA analyser”, is considered the reference method thanks to its ability to determine all AAs in a single analysis, along with automation and good analytical performance (sensitivity, precision, accuracy) [11]. On the other hand, when analyzing very complex matrices the presence of specific interferences requires for tailored separation conditions that cannot be easily optimized by means of the AA analyser. Alternative methods include gas chromatography with flame ionization or mass spectrometry detection (GC-FID or GC-MS) [12,13,14,15] and high performance liquid chromatography (HPLC) with photodiode array (DAD) or fluorescence (FLD) detection [16,17,18,19]. The latter approaches remain the most convenient and widespread analytical methods in AA quantitation; however, a critical optimization of the separation conditions is necessary to achieve adequate resolution of the analytes with respect to the interferences caused by the components of complex biological matrices. 

Importantly, the lack of significant chromophores/fluorophores in most of the AAs requires a suitable derivatization step to allow for UV or FLD detection. Performing pre-chromatographic derivatization can also contribute to reducing AA polarity, thus improving separation. Several derivatization reagents have been used in combination with HPLC separation such as o-phthalaldehyde (OPA), phenyl isothiocyanate (PITC), 3-aminopyridyl-N-hydroxysuccinimidyl carbamate and 9-Fluorenylmethyl chloroformate (FMOC-Cl) [20,21,22,23,24,25].

Many analytical methods based on HPLC-DAD, HPLC-FLD [16,17,18,26,27] or LC-MS [1,6] after derivatization have been proposed for the determination of FAAs in plasma. In this matrix, which contains relatively high levels of analytes and low levels of interferents, up to 20 FAAs have been quantified. In matrices such as milk, colostrum and tissues, characterized by lower FAA level and greater complexity, worse resolutions were obtained, limiting the number of detected/quantified analytes [5,28,29,30]. Reversed phase chromatography (RP-HPLC) with pre-column derivatization [31,32,33] often requires tailored sample pre-treatment to limit matrix interferences. Zanker et al. [26] reported the determination of 20 FAAs derivatized with PITC in colostrum and milk using RP-HPLC after casein coagulation (removed via centrifugation upon addition of CaCl_2_ and chymosin). Wu and Knabe [34] analyzed 19 FAAs and protein-AAs in sow milk and colostrum using HPLC-FLD (derivatization with OPA), except for cysteine (Cys), which was lost through degradation. The determination of proline (Pro) in colostrum and milk has been reported to be difficult because it is highly affected by the presence of matrix interferences. In the study by Wu and Knabe [34], Pro was determined after oxidation in the presence of chloramine-T and NaBH_4_ to 4-amino-l-butanol, which reacted to form a highly fluorescent OPA derivative.

More recently, liquid chromatography–tandem mass spectrometry (LC-MS/MS) has been applied to analyze FAAs in different biological matrices, including plasma, milk and colostrum [35,36,37]. Because of the high selectivity of tandem MS, especially when combined with high-resolution analysis (HRMS) [1], the AAs may not be derivatized and their determination occurs even in a co-elution condition with advantageous short run time. On the other hand, isobaric compounds, such as Leu and isoleucine (Ile), cannot be individually determined with this approach; hence, to achieve baseline separation while avoiding derivatization, two columns are needed, increasing the instrument complexity. Quantitative analysis was also reported from using the isotopic dilution method [38].

Despite the efforts made in the last decades for the optimization of analytical methods for the determination of both FAAs and protein-AAs, their reliable quantitation in complex matrices remains a challenge. 

In the present study, we developed two complementary chromatographic methods for the analysis of FAAs in swine plasma, milk and colostrum. The separation of all 20 FAAs common to all life forms was performed by using two different analytical columns based on core-shell technology, i.e., Kinetex C18 and Kinetex F5, and two detection systems, i.e., DAD and FLD, after pre-column derivatization with FMOC-Cl. Sample de-proteinization in acidic solution and derivatization occurred before the analysis. The analytical method was optimized and validated for specificity, linearity, sensitivity, accuracy and precision. The evaluation of the different combinations of chromatographic columns and detection (DAD and FLD) revealed that the same interfering compounds affected the analyte resolution differently. Hence, to overcome this problem, especially for the more complex milk and colostrum matrices, a combination of both methods was necessary for the quantification of Pro together with the other 19 FAAs. 

## 2. Results

### 2.1. Chromatographic Conditions Optimization

To optimize AA separation, a series of iterated analyses was performed on standard aqueous solutions (each amino acid at a concentration of 3.0 µg/mL) and on real sample solutions.

Amino acid analysis via reversed-phase HPLC is usually performed using C18 columns [1,16,18,19,33,34]. In the present study, three different columns—a Max-RP Synergy (C12), a Kinetex C18 and a Kinetex F5 (pentafluorophenylpropyl)—and two gradient elution programs (Table 1) were tested. Overall, the C18 and the F5 columns provided better peak shapes and separation. The effect of the mobile phase composition was evaluated by considering that gradient elution is often reported in the literature in methods where the aqueous components are buffers at neutral or basic pH: the employment of either sodium acetate buffer [18,32], ammonium acetate buffer [1] or phosphate buffer is reported [16]. The organic modifier of the mobile phase always consists of acetonitrile, sometimes containing a further component such as tetrahydrofuran [32] or methanol [16]. In the present study, acetate and phosphate buffers were tested at different pH values in the range 7.0–8.5, along with different acetonitrile:methanol ratios. At the higher pH values, a better separation was achieved; however, in both of the columns (C18 and F5), reducing the pH of the organic phase in the second part of the gradient was necessary to achieve the separation of histidine (His) and lysine (Lys) peaks, albeit only partially.

The best resolution of the 20 AAs in standard solution was obtained with the F5 column and gradient 1 in 80 min; on the other hand, incomplete separation was achieved with the C18 column and gradient 2 in 40 min (chromatograms in Figure 1).

In the analysis of real samples, matrices of different origin showed different behavior in term of interferences; thus, the chromatographic separations were selected accordingly. The plasma matrix was characterized by interferences that were more easily resolved from the FAAs, especially when using the F5 column and gradient 1 (Figure 1a), compared to milk and colostrum. In the latter two matrices, the determination of Pro was found to be difficult due to the co-elution of interferences, as also reported in the literature [35]. In the present study, separation from the interferences was achieved using method 2 (Figure 1b), which, however, was less selective for a number of other AAs, making this method complementary to method 1 for the determination of Pro. Hence, the determinations of all 20 FAAs in milk and colostrum were achieved by the combination of the two chromatographic methods.

A further aspect that we considered when using FMOC-Cl as the derivatization reagent was the fluorescence quenching of tryptophan (Trp) requiring the application of UV detection to achieve adequate sensitivity.

In summary, in plasma all of the 20 AAs were determined via chromatographic method 1 coupled with FLD/DAD (for Trp) detection. In milk and colostrum, the same analytical method allowed for the determination of 19 AAs; quantification of Pro required the combination of chromatographic method 2 and FLD. A schematic workflow is reported in Figure 2.

### 2.2. Optimization of the Derivatization Reaction

The derivatization reaction was optimized on standard solutions with a total AA concentration of about 3 µg/mL by carrying out experiments using FMOC-Cl 10 mM, 6 mM and 4 mM at room temperature [25]; because no significant differences were observed with regards to the peak areas of the different analytes, the lower FMOC-Cl concentration was selected as the optimum to limit interferences by the reagent excess and the related degradation products. Derivatization time was also optimized by stopping the reaction by addition of 2% formic acid solution after either 8, 10 or 15 min. The obtained responses (derivatives peak area) were not significantly different when stopping the reaction after 10 and 15 min; comparatively, when carrying out the reaction for 8 min, a smaller analytes response was observed, together with a large FMOC-Cl derivatives peak, thus indicating an incomplete derivatization reaction. Finally, 10 min was chosen as the optimum reaction time.

### 2.3. Method Validation

Both chromatographic methods with DAD and FLD detection were validated for specificity, linearity, sensitivity (LOD and LOQ), accuracy, precision (intra-day and inter-day) and matrix effect.

Analytes resolution, as an effective measure of specificity, was established by the injection of single standards and mixtures thereof to determine the retention times of each of the AAs. The obtained chromatograms showed an optimal separation for almost all AAs with both chromatographic systems; the most critical separations were those between His and Lys (method 1 and 2) and between asparagine (Asn) and arginine (Arg) (method 2); nevertheless, accurate quantification was still possible (Figure 1).

Analyte separations obtained in standard solutions were compared to fortified samples with known amounts of solutes. These comparisons showed that the matrix effect depended on the type of sample, i.e., plasma, colostrum or milk (chromatograms in Appendix A). Method 1 was less affected by interferences for all AAs with the exception of Pro, which was then quantified using chromatographic method 2 in the more complex milk and colostrum samples (see Section 2.1). The relative matrix effect (ME%) for all analytes in each sample, evaluated as described in Section 4.7, is reported in Appendix A. ME% values were considered negligible, as they were always in a range of −15.0 to +5.6% for all of the studied analytes.

Due to a higher sensitivity compared to DAD, the quantitative analysis of the biological samples and method validation were conducted with FLD for all FAAs, with the exception of Trp, which was detected at 262 nm.

For all the considered AAs, calibration curves in standard solution were constructed in the range of 0.07–60.0 µg/mL. The peak area of each analyte was plotted against its concentration and linearity parameters were obtained through a linear least-square regression analysis (Appendix A). Determination coefficient values (R^2^) were ≥0.995 for all the compounds, showing linearity over two orders of concentrations. According to the very limited ME% (below ± 15), linearity and sensitivity were determined in the standard solutions. The LOD and LOQ were evaluated by progressive dilution of the standard solutions: the LOD value was established as the signal-to-noise ratio (S/N) = 3, and the LOQ was S/N = 10, where the noise was assumed as the distribution of the response at zero analyte concentration. The obtained LOD and LOQ values ranged from 3 ng/mL to 15 ng/mL and from 10 ng/mL to 50 ng/mL, respectively, for all FAAs. Detailed data are shown in Appendix A.

Accuracy was calculated as recovery %. For each of the samples, recovery experiments were performed by spiking standard amino acid solutions at three known concentrations (low 0.07, medium 0.7 and high 14.0 μg/mL) into plasma, milk and colostrum. The spiked samples were processed as described in Section 4.4 and Section 4.5.

Recoveries were from 70.4 to 114.4%. The low values related to Cys recovery could be reasonably ascribed to partial oxidation or reaction with other compounds of the matrix (Appendix A).

Repeatability (intra-day and inter-day precision), expressed as coefficient of variation (CV%), was determined by analyzing each sample three times on the same day (intra-day, *n* = 3), and over three consecutive days (inter-day, *n* = 9). The results are shown in Appendix A.

### 2.4. Application to Real Samples

Each sample was processed three times as reported in Section 4.4 and Section 4.5 and analyzed via the HPLC method 1 (except for Pro which was quantified with method 2 in milk and colostrum) with FLD/DAD. As expected, the AA content in plasma was higher than that in colostrum or milk (Appendix A and Figure 3), whose analyses were more difficult due to specific interferents and low AA levels.

While the determination of all 20 major FAAs in plasma was achieved using a single chromatographic method (F5 column combined with gradient 1, Figure 1), in milk and colostrum the combination of two chromatographic systems was necessary. Indeed, the use of the F5 column allowed the quantitation of all FAAs with the exception of Pro, whose signal was overlapped by FMOC-Cl excess.

Regarding the comparison of the two detection systems, the sensitivity of FLD was ten-fold that of DAD, and was chosen for the quantitation of all AAs except for Trp, for which DAD was used.

## 3. Discussion

The measurement of FAAs and protein-AAs in biofluids is of utmost importance for nutritional, bio-chemical and metabolic studies, with a fundamental role in the diagnosis and monitoring of inherited disorders of metabolism [39]. Determining the level of FAAs in biofluids is also important for monitoring the effectiveness of dietary interventions in livestock animals [2,3].

Despite the numerous analytical methods reported in the literature addressing the determination of both FFAs and protein-bound AAS, the chromatographic separation and determination of FAAs in complex matrices is still a challenge. In most of the recently published studies [30,38,40,41,42,43,44] reported in Appendix A, importance has been given to the general impact of the AA content/profile on the physio-pathological status of livestock animals, often in relation to nutritional intervention. However, in most of the cited papers, the analytical figures of merit of the applied methods are not considered/reported. Although the lack of this information does not necessarily mean that the cited methods have not been fully validated, our study, by showing in detail the analytical performance of its methods, unambiguously shows the applicability potential.

Here we propose a fully validated HPLC-FLD-DAD method, an alternative to the automatic AA analyser based on ion-exchange chromatography (IEC) with post-column derivatization and UV detection, for the quantitation of FAAs in plasma, milk and colostrum.

The determination of 20 FAAs in biological matrices was carried out by simple de-proteinization and derivatization with FMOC-Cl without the other preparation/clean-up steps involved in previous studies [26]. The combination of two separative methods also allowed for the usually problematic quantification of Pro, avoiding further reaction steps [34].

Thanks to the sensitivity of fluorescence detection (10 times higher than DAD), FAAs at low concentrations were also quantified in complex matrices such as milk and colostrum, as well as in plasma.

Furthermore, while all 19 FAAs were detected with FLD, the possibility of having two detection systems online made it possible to overcome the problem of Trp’s intrinsic fluorescence quenching by quantifying this AA with DAD in a single chromatographic run.

It is generally known that protein-AA concentrations change from colostrum to mature milk in swine as well as in humans and bovines. In fact, Glu, Pro, Met, Ile and Lys have been found to increase with the lactation stage while Cys, Gly, Ser, Thr and Ala have been found to decrease with the lactation stage [10,45,46]. However, less is known regarding the concentration of FAAs in colostrum and milk. In agreement with the present study, Wu and Knabe [34] reported a lower abundance of FAAs in colostrum than in milk.

The highest FAAs observed in milk were Gln, Glu and Gly according to previous reports [9,47], and overall the levels of Ala, Asn, Gln, Glu, Gly, Ile, Leu, Met, Ser, Thr and Tyr were in agreement with the literature [9,47]. Meanwhile, in the present study the concentrations of Arg, Lys and Try were higher and the concentrations of Cys and Pro were found to be lower in comparison with previous reports [9,47]. In milk samples, the observed differences might be due to animal breeds, the stage of lactation and dietary composition [9]. For instance, Lys is the first- or second-limiting AA for milk protein synthesis, especially when corn and soybean meal are used as the primary protein sources in the swine diet [48]. The higher concentration of Arg, coupled with the lower abundance of Pro observed in the milk samples analyzed in the present study, suggests that lower catabolism of Arg via the arginase pathway occurred in the mammary tissue of the sows included in the study [49].

As previously mentioned, information regarding the FAA composition in colostrum is very scarce and dated; however, comparing the data obtained in the present study with the literature, it can be confirmed that Gly, Ala and Glu are the most abundant FAAs in colostrum [34]. The concentrations of Asn, Leu, Lys, Met, Phe and Trp found in our study agree with the previous findings of Wu and Knabe [34]. Generally, the FAA level in colostrum was found to be in line with the content reported in the literature related to the first days of lactation [5,26,30,34,38].

According to Bengtsson [50], the correlation between colostrum and/or milk FAAs and their change in the plasma of piglets receiving them is demonstrated. The more represented FAAs were Ala, Val, Lys, Gln, Gly and Pro, in agreement with previous studies [6,51,52]. In particular, Ala, Arg and Gly were higher with respect to Bertocchi et al. [6] and Yang et al. [52], but agreed with Yu et al. [51]. In conclusion, the proposed method can be very useful to assess the FAA profile in the considered bio-fluids, which has been demonstrated to be strictly related to food intake with regards to dietary intervention in livestock animals.

## 4. Materials and Methods

### 4.1. Chemicals and Solutions

L-Glutamic acid (Glu), L-Aspartic acid (Asp), L-Glutamine (Gln), L-Asparagine (Asn), L-Leucine (Leu), L-Isoleucine (Ile), L-Alanine (Ala), L-Valine (Val), L-Tyrosine (Tyr), L-Threonine (Thr), L-Serine (Ser), L-Arginine (Arg), L-Methionine (Met) and L-Cysteine (Cys) were purchased from Fluka. L-Glycine (Gly), L-Tryptophan (Trp), L-Histidine (His) and L-Lysine (Lys) were purchased from Sigma-Aldrich. L-Phenylalanine (Phe) was obtained from Carlo Erba Reagents (Cornaredo, Italy). All amino acids were commercialized, pure and pharma grade (purity at least ≥ 99%), except the Lys standard, which was given as L-Lysine monohydrochloride (minimum 98% pure). The 9-Fluorenylmethyl chloroformate (FMOC-Cl) was from Sigma-Aldrich.

HPLC-grade acetonitrile and methanol were purchased from Merck. Sodium tetraborate, potassium carbonate and sodium mono- and dihydrogen phosphate, formic acid (99%) and perchloric acid (70%) were purchased from Sigma-Aldrich. Purified water produced with a Milli-RX apparatus (Millipore, Milford, MA, USA) was employed for the preparation of all solutions, buffers and mobile phase.

Max-RP Synergy (C12), Kinetex C18 and Kinetex F5 columns for HPLC were purchased from Phenomenex (Torrance, CA, USA).

### 4.2. Sample Collection

Colostrum and milk samples were collected from 10 sows at farrowing (colostrum), and at day 20 (Milk d20) of lactation. Farrowing was not induced, and colostrum was collected in the period between the birth of the first piglet and the delivery of the last one. Samples were collected across all of the sows’ teats as reported by Luise et al. [53]. From each sample, an aliquot of colostrum and milk was snap frozen and then preserved at −80 °C. Blood samples were collected from 10 weaned piglets from the same sows via jugular venipuncture into K3 EDTA tubes (Vacutest Kima Srl, Arzergrande PD Italy). Tubes were immediately centrifuged at 4 °C at 4000× *g* for 10 min to obtain plasma that was than collected into a sterile tube, snap-frozen in liquid nitrogen and then preserved at −80 °C. All samples were collected from healthy animals reared in the same commercial farm and fed the same diet.

The sample collection was conducted in a commercial multiplication unit located in the so-called “Italian Food Valley”. The animals involved in the present study were sows and piglets subjected to conventional farm rearing conditions in Europe (EU) as according to the Dir. 120/2008 EC.

All collected samples were thawed and then pooled in order to provide a pool for each single matrix with a total amount of 15 mL. Each pool was then preserved at −80 °C until AA analysis.

### 4.3. Standard Solutions

Asp, Glu, Asn, Gln, Ser, Gly, Thr, Arg, Ala, Pro, Val, Met, Ile, Leu, Phe, Trp, His, Lys, Cys and Tyr were accurately weighed and dissolved in deionized water to prepare a stock solution (1 mg/mL). Via serial dilution, five solutions were prepared at the following concentrations for each of the considered AAs: 0.07, 0.30, 0.7, 7.0, 14.0 and 60.0 µg/mL.

### 4.4. Sample Preparation

An aliquot of 1 mL of each pool (milk, colostrum, plasma) was centrifuged at 12,300 rcf for 10 min. The supernatant was separated and mixed with the same volume of perchloric acid (1.5 M) for protein precipitation (5 min at room temperature). The mixture was centrifuged at 12,300 rcf for 5 min and the supernatant (4 parts) was neutralized by the addition of aqueous K_2_CO_3_ 2M (1 part); the obtained mixture was centrifuged at 12,300× *g* for 5 min and the supernatant was subjected to derivatization reaction.

### 4.5. Derivatization Reaction

A 100 µL aliquot of treated sample solution or standard solution was mixed with 100 µL of sodium tetraborate buffer (0.1 M, pH 8.5); the buffered solution was mixed with 200 µL of 4 mM solution of FMOC-Cl in acetonitrile. The obtained clear solution was kept at room temperature for 10 min and the derivatization reaction was stopped by the addition of 150 µL of HCOOH 1% (aqueous solution). The solution was injected into the HPLC system.

### 4.6. HPLC-UV/FLD Conditions

A Jasco Model LG-980-02S ternary gradient unit, a HP1050 diode array detector (DAD) and a Jasco FP-920 fluorescence detector (FLD) (Jasco Corporation, Tokyo, Japan) were used. HPLC method 1 was performed at room temperature by using a Kinetex F5 (5 µm, 150 × 4.6 mm i.d., Phenomenex) column; the mobile phase constituted of 40 mM aqueous NaH_2_PO_4_ buffer, pH 8.5 (component A), and a solution made up of CH_3_CN (45%), CH_3_OH (45%), H_2_O (10%) and H_3_PO_4_ (0.1%) (component B). Gradient elution (gradient 1) conditions were applied as reported in Table 1 at a flow rate of 1 mL/min, and the injection volume was 50 µL. After each run, a 5 min reconditioning step was performed at the initial conditions. HPLC method 2 was performed with slight modifications to the conditions reported by Themelis et al. [54]. Briefly, the separations were carried out at room temperature (25 °C) on a Phenomenex Kinetex core-shell 5 μm C18 (150 × 4.6 mm i.d.) column with a mobile phase consisting of ammonium formate 60 mM, pH 5.5 (A), ammonium formate 60 mM, pH 7.5 (B), and ammonium formate 10 mM and formic acid 0.1% in ACN/water, 90:10 (*v/v*) (C). A stepwise gradient elution (gradient 2 in Table 1) was applied. After each run, a 5 min reconditioning step was performed at the initial conditions. The flow rate was set at 1.2 mL/min and the injection volume was 20 μL.

The two HPLC methods were combined with two on-line detection systems: UV detection at 262 nm and FL detection at λ_exc_ = 265 nm and λ_em_ = 315 nm.

### 4.7. Method Validation

Analytical methods for the determination of AAs in plasma, colostrum and milk samples were validated according to ICH guidelines [55].

Selectivity was evaluated by comparing the chromatograms obtained from standard, sample and spiked sample solutions.

The solutions of the derivatized AAs were prepared in triplicate, obtaining the following final concentrations of 0.07, 0.30, 0.7, 7.0, 14.0 and 60.0 µg/mL for linearity studies. These solutions were analyzed with the two HPLC-DAD-FLD methods as described above and calibration graphs were constructed by plotting the peak area versus the corresponding AA concentration.

The limit of detection (LOD) and the limit of quantitation (LOQ), calculated as a signal-to-noise (S/N) ratio of 3 and 10, respectively, for each analyte, were determined by injecting step-wise dilutions of the standard solutions.

The matrix effect was evaluated at three concentration levels (0.07, 0.7 and 14.0 µg/mL) by comparing the standard areas obtained from standard solutions and the spiked samples after pre-treatment and before derivatization. The absolute matrix effect was calculated as the difference between the peak area of the spiked sample and the peak area of the standard solution. The relative matrix effect for all analytes was evaluated as the ratio between the absolute matrix effect and the peak area of standard solutions, expressed as percentage. A T-test was used to compare the calibration curve slopes (*p* < 0.05).

Accuracy was determined as the recovery % on spiked samples at three concentration levels—low (0.07 μg/mL), medium (0.7 μg/mL) and high (14.0 μg/mL)—by comparing the analyte peak areas of samples fortified before treatment and those of the standard solutions.

The endogenous contributions of naturally occurring analytes in plasma, colostrum and milk were subtracted from the analysis of spiked samples during the evaluation of recovery and matrix effects.

Precision (intra and inter-day) was evaluated via triplicate analysis over three different days (n = 9), by analyzing standard solutions at low (0.07 μg/mL), medium (0.7 μg/mL) and high (14.0 μg/mL) concentration levels.

## Figures and Tables

**Figure 1 molecules-27-04153-f001:**
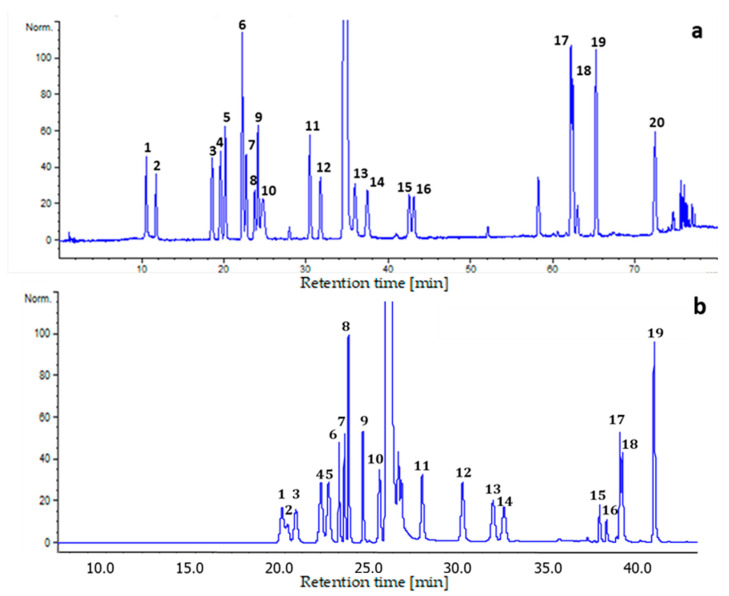
Chromatograms of standard solutions. (**a**) Method 1, mobile phase A: 40 mM aqueous NaH_2_PO_4_ buffer, pH 8.5, B: CH_3_CN/CH_3_OH/H_2_O/H_3_PO_4_ 45:45:10:0.1 (*v/v*); gradient 1; Kinetex F5 column. Detection: DAD at 262 nm. FAAs: 1-Asp, 2-Glu, 3-Asn, 4-Gln, 5-Ser, 6-Gly, 7-Thr, 8-Arg, 9-Ala, 10-Pro, 11-Val, 12-Met, 13-Ile, 14-Leu, 15-Phe, 16-Trp, 17-His, 18-Lys, 19-Cys, 20-Tyr. (**b**) Method 2, mobile phase A: ammonium formate 60 mM, pH 5.5, B: ammonium formate 60 mM, pH 7.5, C: ammonium formate 10 mM and formic acid 0.1% in ACN/water, 90:10 (*v/v*); gradient 2; Kinetex core-shell C18 column. Detection: FLD, λ_exc_ = 265 nm and λ_em_ = 315 nm. FAAs: 1-Asn, 2-Arg, 3-Gln, 4-Ser, 5-Asp, 6-Glu, 7-Thr, 8-Gly, 9-Ala, 10-Pro, 11-Met, 12-Val, 13-Ile, 14-Leu, 15- Phe, 16-Cys, 17-His, 18-Lys, 19-Tyr. For abbreviation meanings see Section 4.1.

**Figure 2 molecules-27-04153-f002:**
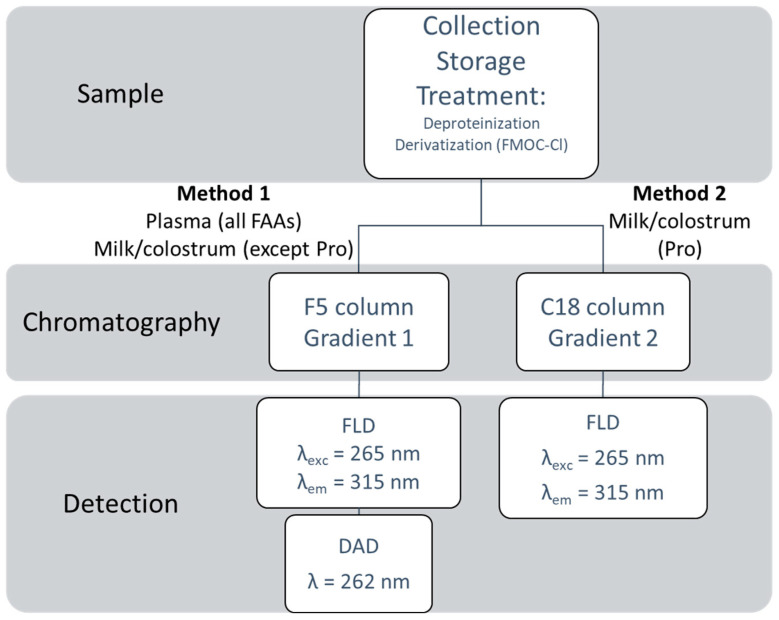
Workflow for the analysis of the 20 FAAs in milk, colostrum and plasma.

**Figure 3 molecules-27-04153-f003:**
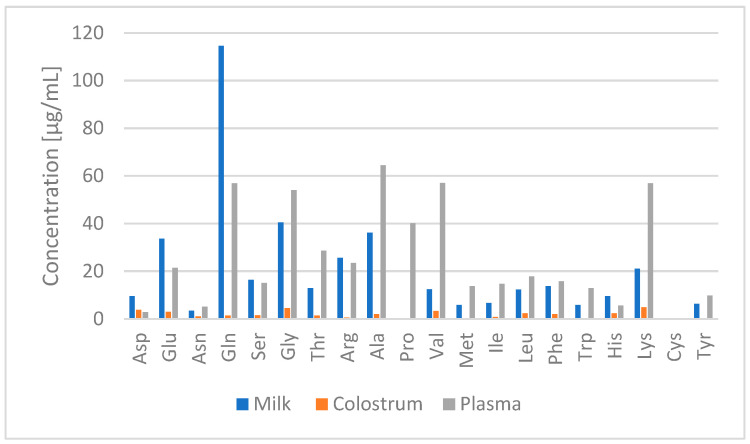
Concentrations of free AAs in plasma, colostrum and milk. The data are the mean of three replicated experiments (CV% ≤ 9.5).

**Table 1 molecules-27-04153-t001:** HPLC gradient conditions of method 1 (gradient 1) and method 2 (gradient 2).

Gradient 1 *			Gradient 2 ^⊥^			
Time (min)	A%	B%	Time (min)	A%	B%	C%
0	80	20	0	90	0	10
28	57	43	10	85	0	15
38	57	43	11	80	0	20
70	25	75	20	76	0	24
75	0	100	21	0	55	45
			31	0	55	45
			36	0	30	70
			42	0	15	85

* A: 40 mM aqueous NaH_2_PO_4_ buffer, pH 8.5; B: CH_3_CN/CH_3_OH/H_2_O/H_3_PO_4_ 45:45:10:0.1 (*v/v*). Kinetex F5 column. ^⊥^ A: ammonium formate 60 mM, pH 5.5; B: ammonium formate 60 mM, pH 7.5; C: ammonium formate 10 mM and formic acid 0.1% in ACN/water, 90:10 (*v/v*). Kinetex core-shell C18 column.

## Data Availability

Not applicable.

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
