# Peer review of "Determination of Free Amino Acids in Milk, Colostrum and Plasma of Swine via Liquid Chromatography with Fluorescence and UV Detection"

_molecules, 2022, doi:10.3390/molecules27134153_

Round 1
Reviewer 1 Report
In this manuscript, the authors developed two complementary chromatographic methods for the analysis of FAAs in swine plasma, milk and colostrum. The manuscript was showing some interesting results for further improvement in the FAAs analysis and qualify for publication in the Molecules. My detailed comments are as follows:
1. There is no novelty comparison in the manuscript reported. All the recent literature (2020-2022) needs to be compared and tabulated.
2. On lines 117, authors mentioned “many methods reported in literature have not been fully validated”, but the authors did not provide the reference and detail invalid aspects. Please provide more information and data.
3. On line 236, authors proposed the relative matrix effect (ME%). How to carried out the experiment and calculate the values?
4. Application to real samples Section, why were concentration of the free AAs in colostrum so low, and whether did the results compare with that of standard methods.
5. On line 176, please check subscript of NaH2PO4.
Author Response
Reviewer 1
In this manuscript, the authors developed two complementary chromatographic methods for the analysis of FAAs in swine plasma, milk and colostrum. The manuscript was showing some interesting results for further improvement in the FAAs analysis and qualify for publication in the Molecules. My detailed comments are as follows:
- There is no novelty comparison in the manuscript reported. All the recent literature (2020-2022) needs to be compared and tabulated.
R1. See response at point 2 below.
- On lines 117, authors mentioned “many methods reported in literature have not been fully validated”, but the authors did not provide the reference and detail invalid aspects. Please provide more information and data.
R2. The sentence has been removed and the information required has been detailed in the Discussion section (lines 328-336) also in relation to a newly added table (Table S7, in supplementary material) as requested at point 1 above. Five additional references (2020-2022) have been introduced and discussed.
- On line 236, authors proposed the relative matrix effect (ME%). How to carry out the experiment and calculate the values?
R3. Details on ME% experiments and calculation are reported in Section 4.7. At lines 480-486, we have added further information to the better comprehension of the following sentence at lines 490-492 (the peak area of spiked samples was normalized when naturally occurring AAs are contained).
- Application to real samples Section, why were concentration of the free AAs in colostrum so low, and whether did the results compare with that of standard methods.
R4. The level of FAAs in colostrum span in wide range depending on the day of lactation according to the literature. The values found in our study are in line with those of references 5, 26, 30, 34, 38. In particular, a recently published paper (2020 ref 30) reports very similar values in sow colostrum as quantified by the standard method (AA analyser). A sentence was added in the Discussion section (Lines 372-374).
- On line 176, please check subscript of NaH2PO4.
R5. Done.
Reviewer 2 Report
The manuscript concerns an analytical method for determining free amino acids in milk, colostrum and plasma by a HPLC-DAD and FLD method. The approach is well described and the results very interesting.
Some comments:
Page 2 , line 54: In this sentence it seems something is missing, maybe it should be related to the following sentence.
Page 2, line 94: this sentence in not very clear. The author should try to modify it.
Page 6, line 225: Is this defenition of seletivity right? The authors should clarify this concept
Author Response
The manuscript concerns an analytical method for determining free amino acids in milk, colostrum and plasma by a HPLC-DAD and FLD method. The approach is well described and the results very interesting.
Some comments:
Page 2, line 54: In this sentence it seems something is missing, maybe it should be related to the following sentence.
- The sentence has been corrected.
Page 2, line 94: this sentence in not very clear. The author should try to modify it.
- The sentence has been modified.
Page 6, line 225: Is this defenition of seletivity right? The authors should clarify this concept
- The sentence has been improved.
Reviewer 3 Report
The authors propose a new analytical method for the quantification of free amino acids in three organic matrices of animal origin. Two chromatographic methods, diode array detector (DAD) and fluorescence detector (FLD), were compared, demonstrating that for the quantification of the totality of free amino acids it is necessary to complement the two methods.
The authors present a good introduction, contextualize the research topic, clearly present the knowledge gap, and accurately and clearly present the purpose of the research. The methodology is explicit and clearly written, which allows the replication of the study.
The results obtained allow the development of a solid discussion, which is based on studies presented by other authors.
Finally, it can be established that the analytical method presented by the authors is a novel alternative that allows the quantification of free amino acids in organic matrices such as milk, colostrum and plasma.
Author Response
Thanks for your comment.
Reviewer 4 Report
Journal: Molecules
Manuscript ID: molecules-1771522
Determination of free amino acids in milk, colostrum and plasma of swine by liquid chromatography with fluorescence and UV detection
Roberto Gotti , Erika Esposito , Diana Luise , Stefano Tullio , Nicolò Interino , Paolo Trevisi , Jessica Fiori.
Abstract: Amino acids are ubiquitous components of mammalian milk and highly contribute to its nutritional value. The compositional analysis of free amino acids is poorly reported in the literature even though their determination in the biological fluids of livestock animals is necessary to establish possible nutritional interventions. In the present study, the free amino acids profile in swine mature milk, colostrum and plasma was assessed by targeted metabolomics approach. In particular, 20 amino acids were identified and quantified by two alternative and complementary reversed-phase HPLC methods, involving two stationary phases based on core-shell technology, i.e., Kinetex C18 and Kinetex F5, and two detection systems, diode array detector (DAD) and fluorescence detector (FLD). The sample preparation involved a de-proteinization step, followed by pre-chromatographic derivatization with 9-fluorenylmethylchloroformate (FMOC-Cl). The two optimized methods were validated for selectivity, linearity, sensitivity, matrix effect, accuracy and precision and the analytical performances were compared. The analytical methods proved to be suitable for the free amino acids profiling in different matrices with a high sensitivity and specificity. The correlation among the amino acids levels in the different biological fluids can be useful for the evaluation of physio- pathological status and to monitor the effects of therapeutic or nutritional interventions in humans and animals.
Comments:
It is a topic of interest to the researchers in the related area but the paper needs minor improvements before acceptance for publication. My detailed comments are as follows:
The materials and methods in the paper works very well (Sample collection, Standard solutions, Sample preparation, Derivatization reaction and HPLC-UV/FLD conditions), especially the part that correspond to HPLC-UV/FLD conditions.
The proposed method can be very useful to assess the FAA profile in the considered bio-fluids (and perhaps for anothers…), that has demonstrated to be strictly related to the food intake addressing the dietary intervention in livestock animals. In the technical aspects of the work, it is well described and well executed. There should not be any inconvenience when carrying out this work in another laboratory.
I have not detected any excess of citations of the authors in the manuscript, knowing that the authors work in the research field.
As a suggestion, It would seem convenient to use in the discussion some additional bibliographic support that will support the importance of the FAA profile in the considered bio-fluids described. I believe that these references would add a more realistic and verified scenario to the discussion and interpretation of the results obtained.
I consider, once again as I have mentioned previously, that a few minimal interventions and contributions are necessary to increase the quality (already good) of the work (method).
Author Response
It is a topic of interest to the researchers in the related area but the paper needs minor improvements before acceptance for publication. My detailed comments are as follows:
The materials and methods in the paper works very well (Sample collection, Standard solutions, Sample preparation, Derivatization reaction and HPLC-UV/FLD conditions), especially the part that correspond to HPLC-UV/FLD conditions.
The proposed method can be very useful to assess the FAA profile in the considered bio-fluids (and perhaps for anothers…), that has demonstrated to be strictly related to the food intake addressing the dietary intervention in livestock animals. In the technical aspects of the work, it is well described and well executed. There should not be any inconvenience when carrying out this work in another laboratory.
I have not detected any excess of citations of the authors in the manuscript, knowing that the authors work in the research field.
As a suggestion, It would seem convenient to use in the discussion some additional bibliographic support that will support the importance of the FAA profile in the considered bio-fluids described. I believe that these references would add a more realistic and verified scenario to the discussion and interpretation of the results obtained.
- Some additional bibliographic supports have been introduced and discussed. In particular, as requested the importance of FAA profile have been underlined in the discussion section (lines 320-324) to further support what it has been already reported in the original submission (Introduction section).
I consider, once again as I have mentioned previously, that a few minimal interventions and contributions are necessary to increase the quality (already good) of the work (method).